Taphonomy and taxonomy of a juvenile lambeosaurine (Ornithischia: Hadrosauridae) bonebed from the late Campanian Wapiti Formation of northwestern Alberta, Canada

Holland Brayden 1 blongle2@myune.edu.au
http://orcid.org/0000-0001-5890-8183 Bell Phil R. 1
Fanti Federico 2
Hamilton Samantha M. 3
Larson Derek W. 4
Sissons Robin 3
http://orcid.org/0000-0002-5488-6797 Sullivan Corwin 3 4
Vavrek Matthew J. 5
Wang Yanyin 3
http://orcid.org/0000-0002-4205-9794 Campione Nicolás E. 1 ncampion@une.edu.au
1 Palaeoscience Research Centre, School of Environmental and Rural Science, University of New England , Armidale, New South Wales , Australia
2 Dipartimento di Scienze Biologiche, Geologiche e Ambientali, Alma Mater Studiorum, Università di Bologna , Bologna , Italy
3 Department of Biological Sciences, University of Alberta , Edmonton, Alberta , Canada
4 Philip J. Currie Dinosaur Museum , Wembley, Alberta , Canada
5 Department of Natural History, Royal Ontario Museum , Toronto, Ontario , Canada
Farke Andrew
Electronic publication date: 2021 May 4
Publication date: 2021
Volume: 9
Electronic Location ID: e11290
Received 2020 Dec 8; Accepted 2021 Mar 26
Copyright: © 2021 Holland et al.
Copyright year: 2021
Copyright holder: Holland et al.
License: This is an open access article distributed under the terms of the Creative Commons Attribution License, which permits unrestricted use, distribution, reproduction and adaptation in any medium and for any purpose provided that it is properly attributed. For attribution, the original author(s), title, publication source (PeerJ) and either DOI or URL of the article must be cited.
License URL: https://creativecommons.org/licenses/by/4.0/

Keywords: Bonebed, Age segregation, Campanian, Wapiti Formation, Palaeontology, Ornithischia, Hadrosauridae, Lambeosaurinae, Dinosauria, Taphonomy

Funding: Research Training Program and Rural and Regional Enterprise Dinosaur Research Institute (DRI) Student Project DRI Dinosaur Fieldwork in Western Canada Australian Research Council Discovery Early Career Research Award DE190101423 Natural Sciences and Engineering Research Council Discovery RGPIN-2017-06246 University of Alberta to Corwin Sullivan This research was funded by a Research Training Program and Rural and Regional Enterprise scholarships to Brayden Holland, a Dinosaur Research Institute (DRI) Student Project Grant to Brayden Holland, a DRI Dinosaur Fieldwork in Western Canada to Nicolás E. Campione and Matthew J. Vavrek, an Australian Research Council Discovery Early Career Research Award to Nicolás E. Campione (DE190101423), and a Natural Sciences and Engineering Research Council Discovery Grant (RGPIN-2017-06246) and an endowment associated with the Philip J. Currie Professorship at the University of Alberta to Corwin Sullivan. The funders had no role in study design, data collection and analysis, decision to publish, or preparation of the manuscript.

==============================
Hadrosaurid (duck-billed) dinosaur bonebeds are exceedingly prevalent in upper Cretaceous (Campanian–Maastrichtian) strata from the Midwest of North America (especially Alberta, Canada, and Montana, U.S.A) but are less frequently documented from more northern regions. The Wapiti Formation (Campanian–Maastrichtian) of northwestern Alberta is a largely untapped resource of terrestrial palaeontological information missing from southern Alberta due to the deposition of the marine Bearpaw Formation. In 2018, the Boreal Alberta Dinosaur Project rediscovered the Spring Creek Bonebed, which had been lost since 2002, along the northern bank of the Wapiti River, southwest of Grande Prairie. Earlier excavations and observations of the Spring Creek Bonebed suggested that the site yielded young hadrosaurines. Continued work in 2018 and 2019 recovered ~300 specimens that included a minimum of eight individuals, based on the number of right humeri. The morphology of several recovered cranial elements unequivocally supports lambeosaurine affinities, making the Spring Creek sample the first documented occurrence of lambeosaurines in the Wapiti Formation. The overall size range and histology of the bones found at the site indicate that these animals were uniformly late juveniles, suggesting that age segregation was a life history strategy among hadrosaurids. Given the considerable size attained by the Spring Creek lambeosaurines, they were probably segregated from the breeding population during nesting or caring for young, rather than due to different diet and locomotory requirements. Dynamic aspects of life history, such as age segregation, may well have contributed to the highly diverse and cosmopolitan nature of Late Cretaceous hadrosaurids.

Introduction

Macrofossil bonebeds are sources of palaeontological data that greatly contribute to our understanding of anatomy, diversity, life history, community structure, behaviour, population dynamics and taphonomy (Rogers, Eberth & Fiorillo, 2007). In North America, hadrosaurid dinosaur bonebeds are particularly concentrated in uppermost Cretaceous (Campanian–Maastrichtian) deposits, notably in those of the Belly River and Edmonton groups in southern Alberta, Canada (Getty et al., 1998; Eberth & Getty, 2005; Eberth & Currie, 2010; Bell & Campione, 2014; Eberth, 2015; Evans et al., 2015) and the Two Medicine, Hell Creek, Lance, and Judith River formations in the northern part of the western United States (Christians, 1992; Varricchio & Horner, 1993; Britt et al., 2009; Scherzer & Varricchio, 2010; Keenan & Scannella, 2014; Prieto-Márquez & Gutarra, 2016). Hadrosaurid specimens from bonebeds in these formations were among the first dinosaurs to be histologically sampled, which allowed for the reconstruction of their growth rates (Horner & Currie, 1994; Horner, Ricqles & Padian, 1999; Horner, De Ricqles & Padian, 2000) and provided the first evidence for parental care in dinosaurs (Horner & Makela, 1979; Horner, De Ricqles & Padian, 2000). Despite their frequency and importance, large numbers of North American hadrosaurid bonebeds have not been described in detail, particularly in northern rock units such as the Wapiti Formation (Fanti & Catuneanu, 2009; Fanti & Miyashita, 2009). These offer the opportunity to explore the diversity and preservation of hadrosaurids outside the traditionally sampled North American strata.

In northwestern Alberta, Wapiti Formation deposits span the mid-Campanian to upper Maastrichtian and are contemporaneous with most of the Belly River and Edmonton groups in southern Alberta (Fanti & Catuneanu, 2009; Eberth & Braman, 2012). Unlike its more famous southern counterparts, which are interrupted by marine transgressions of the Bearpaw Formation, the Wapiti Formation is a continuous package of terrestrial sediments (Eberth & Getty, 2005; Fanti & Catuneanu, 2009; Eberth & Braman, 2012). Although the Wapiti Formation was originally only for a single ceratopsian site, the Pipestone Creek Bonebed (Currie, Langston & Tanke, 2008b), fieldwork over the past 10–15 years has uncovered abundant vertebrate ichnofossils (Bell, Fanti & Sissons, 2013; Fanti, Bell & Sissons, 2013), articulated skeletons with skin impressions (Bell et al., 2014a; Bell et al., 2014b; Enriquez et al., 2021; Enriquez et al. in press), microfossil sites (Fanti & Miyashita, 2009), and macrofossil bonebeds (Tanke, 2004; Currie, Langston & Tanke, 2008b; Fanti, Currie & Burns, 2015) (Fig. 1).

Figure 1 Locality map of main macrofossil localities from the Grande Prairie area and the geographic extent of the Wapiti Formation.

(A) Locality map of the main macrofossil localities proximate to Grande Prairie, Alberta, Canada. Numbers indicate the following localities: (1) George Robinson Bonebed (Tanke, 2004); (2) Mummified Edmontosaurus regalis skeleton (Bell et al., 2014a); (3) Red Willow hadrosaur (Bell et al., 2014b); (4) Wapiti River Pachyrhinosaurus Bonebed (Fanti, Currie & Burns, 2015); (5) Pipestone Creek Pachyrhinosaurus lakustai Bonebed (Currie, Langston & Tanke, 2008a); (6) Spring Creek Bonebed (red star; this study). (B) Map illustrating the lateral extent of the Wapiti Formation (in grey) across Alberta and into eastern British Columbia.

The Pipestone Creek Bonebed was discovered in 1974 (Tanke, 2004) and has produced disarticulated bones representing at least 27 juvenile- to adult-sized individuals of the ceratopsian Pachyrhinosaurus lakustai, along with remains of the dromaeosaurid Boreonykus certekorum, tyrannosaurids, and non-dinosaurian vertebrates (Currie, Langston & Tanke, 2008b; Bell & Currie, 2016). Its unique faunal content has been used to support dinosaur endemism hypotheses across Laramidia during the Late Cretaceous (Currie, Langston & Tanke, 2008b; Sampson et al., 2010; Lucas et al., 2016). The Wapiti River Bonebed, a second ceratopsian bonebed located west of the Pipestone Creek Bonebed, is dominated by Pachyrhinosaurus specimens that have not yet been conclusively identified at the species level. Notably, this bonebed represents one of the most inland occurrences of centrosaurine ceratopsians in North America, given its inferred location relative to the Western Interior Seaway (Fanti, Currie & Burns, 2015). In addition to macrofossil bonebeds (defined as >75% of specimens with a preserved length >5 cm; sensu Eberth, Rogers & Fiorillo, 2007), the Kleskun Hill microfossil site (defined as >75% of specimens with a preserved length <5 cm) preserves a high diversity of vertebrates, including fish, lizards, dinosaurs, and mammals (Fanti & Miyashita, 2009). Several additional monodominant hadrosaurid bonebeds have subsequently been discovered, although not yet documented in detail (Tanke, 2004; Bell et al., 2014a; Bell et al., 2014b).

The hadrosaurids of the Wapiti Formation are taxonomically enigmatic. Edmontosaurus regalis is currently the only species reported from this temporally extensive formation (Bell et al., 2014a; Bell et al., 2014b). The majority of hadrosaurid material so far recovered came from Unit 4 of the formation, which is broadly contemporaneous with portions of the Horseshoe Canyon Formation of southern Alberta, from which E. regalis is commonly recovered (Bell et al., 2014a; Bell et al., 2014b; Campione & Evans, 2011; Eberth et al., 2013). Moreover, there is yet to be any definitive evidence to suggest the presence of another hadrosaurid taxon beside E. regalis in Unit 4. However, it is unlikely that E. regalis was the only hadrosaurid from the entire formation, given the known diversity of hadrosaurids elsewhere in Alberta and the temporal extent of the Wapiti Formation. For instance, lambeosaurines have yet to be documented from the formation, despite their ubiquity within both the Belly River and Edmonton groups (Lull & Wright, 1942; Evans, Forster & Reisz, 2005; Ryan & Evans, 2005; Evans & Reisz, 2007; Evans, Reisz & Dupuis, 2007; Evans, 2010; Brink et al., 2011; Mallon et al., 2012; Eberth et al., 2013; Farke et al., 2013).

In 1988, Grande Prairie Regional College staff discovered several well-preserved hadrosaurid bones along the northern bank of the Wapiti River, approximately 150 m downstream of the confluence with the Spring Creek (Tanke, 2004). The site was dubbed the Spring Creek Bonebed (SCBB), and the material was suggested to pertain to Hadrosaurinae, based on the “low deltoid crests” morphology seen in the recovered humeri. Given their size range, the bones were interpreted as the remains of subadult individuals that may have formed a “bachelor herd” (Tanke, 2004). Initial excavations at the SCBB undertaken by Grande Prairie Regional College and Royal Tyrrell Museum began in 1988, resulting in 40 specimens recovered between 1988 and 2002. By 2003, however, the site had been obscured by riverbank slumping (Tanke, 2004) and could not be rediscovered despite repeated attempts over the following years. The bonebed was finally rediscovered in 2018 by one of us (MJV) as part of the Boreal Alberta Dinosaur Project and subsequently excavated during the 2018 and 2019 field seasons (Fig. 2). These recent excavations secured hundreds of new hadrosaurid specimens, including the first diagnostic cranial material.

Figure 2 Quarry map of the Spring Creek Bonebed.

(A) Map of the 2018 and 2019 excavations of the Spring Creek Bonebed by the Boreal Alberta Dinosaur Project (grey: isolated specimens; white: specimens in concretions). (B) Associated dentary (1; UALVP 59898), partial dental battery (2; UALVP 59887), and predentary (3; UALVP 59888) from Spring Creek Bonebed. Reconstruction based on Hypacrosaurus stebingeri (Brink et al., 2011). The 10 cm scale bar applies to the bones in (B) and the skull reconstruction. (C) A rose diagram of the recorded orientations of long bones from the Spring Creek Bonebed showing a preferential NE–SW modality, but overall high circular variance. (D) Quarry photo of bones in situ.

In this study, we describe the anatomy of the most taxonomically informative hadrosaurid bones preserved at the SCBB and examine the taphonomic factors that may have formed the bonebed. We test the original suggestion that the material might belong to Hadrosaurinae (Tanke, 2004) using a larger sample that encompasses more diagnostic elements and use histological analyses to assess the age distribution of the bonebed sample. Finally, we consider the nature of fossil deposition at the SCBB, with a particular focus on whether the bonebed assemblage originated through attrition or mass mortality, and explore its implications for our understanding of hadrosaurid life histories.

Geological Setting

Outcrops of the terrestrial Wapiti Formation are exposed extensively in central to northwestern Alberta and into the eastern-most regions of British Columbia (Fig. 1). Stratigraphically, the Wapiti Formation overlies the marine Puskwaskau Formation and underlies the terrestrial Scollard Formation (Fanti & Catuneanu, 2009). Spanning from the mid-Campanian (~79.1 Ma) into the Maastrichtian (~67 Ma), the Wapiti Formation is roughly contemporaneous with the Belly River and Edmonton groups of southern Alberta and the Two Medicine and St. Mary River formations of northwestern Montana (Fanti & Catuneanu, 2010; Eberth & Kamo, 2020; Zubalich et al., 2021). The formation is subdivided into five units that suggest an overall progression from channel-fill sandstones to floodplain-derived finer sediments (Fanti & Catuneanu, 2009, 2010). Coals from Unit 3 and the Red Willow coal zone (upper Unit 4) are interpreted as synchronous with the maximum flooding surfaces of the marine Bearpaw Formation and Drumheller Marine Tongue, respectively (Fanti & Catuneanu, 2009). However, actual marine sediments do not interrupt the succession of terrestrial strata in this region, as they do in southern Alberta. Therefore, the Wapiti Formation represents a nearly continuous mid-Campanian–Maastrichtian terrestrial record that is important for tracking faunal transformation in northern Laramidia, particularly when marine transgressions inundated southern Alberta.

Because the Wapiti Formation exposures in which the SCBB is located are highly unstable and prone to slumping, the stratigraphic position of the bonebed is limited to within a few metres. The SCBB is located ~11.5 km downstream of the Pipestone Creek Bonebed, placing it within Unit 3 of the Wapiti Formation and implying rough contemporaneity with the lowermost units of the Horseshoe Canyon Formation (the Strathmore and Drumheller members: Currie, Langston & Tanke, 2008a; Eberth & Braman, 2012; Zubalich et al., 2021). Unit 3 comprises channel sandstones overlain by interbedded mudstones and siltstones, minor sandstone sheets, and extensive coals, representing fluvial point bars within high-sinuosity fluvial systems in floodplain environments (Fanti & Catuneanu, 2009). At the SCBB locality (Fig. 3), approximately 14 vertical metres of the Wapiti Formation are exposed on a cut bank of the Wapiti River, where slumping has obscured some sedimentary features and the boundaries between horizons. Nevertheless, massive mudstones up to 5.5 m thick (interrupted by thin sandy layers) dominate the exposure, alternating with sandstones up to 2.8 m thick (Fig. 3). The SCBB is confined to a ~40 cm thick horizon within the middle of a massive, organic-rich mudstone approximately 3.7 m thick. Bones exhibit no signs of grading, range from 10 to 640 mm in maximum length, and have no distinct preferred orientation (see “Results”; Fig. 2C). In addition to bones, coalified plant remains (<10 cm long), clay nodules, and amber (<2 cm long) are also present in the bonebed. Conformably underlying the bonebed-hosting mudstone is a ~80 cm thick sandstone with shallow crossbedding, which overlies a coal layer that is only exposed during low water periods. The overall sedimentary evidence indicates the SCBB was deposited on a vegetated floodplain traversed by the meandering rivers that were the main depositional environment for Unit 3 (Fanti & Catuneanu, 2009).

Figure 3 Exposures at the Spring Creek Bonebed.

(A) Photograph of the bank exposure at the Spring Creek Bonebed (indicated by white arrows). (B) Stratigraphic column from the Spring Creek Bonebed (sediment grains sizes: c, clay; m, mud; fs, fine sand; s, sand). Derek Larson (175 cm) for scale.

Materials & methods

Excavation

Specimens collected by the Grande Prairie Regional College in 1988 and 1991 were mapped but could not be placed in our quarry maps or used in our quarry analyses because accompanying orientation data, field identifications, and field numbers were not recorded. During the Boreal Alberta Dinosaur Project excavations in 2018 and 2019, orientation, plunge, and maximum preserved length were recorded on-site for all specimens with a length:width ratio ≥2. Obvious taphonomic artefacts, such as fracturing, were noted. All specimens with a total length >5 cm were mapped by hand in 1 × 1 m grids, subdivided into 10 × 10 cm squares. Specimens collected in 1988 and 1991 are accessioned at the Royal Tyrrell Museum of Palaeontology (TMP), Drumheller, Alberta, Canada (TMP1988.094 and TMP1991.137 series), whereas specimens collected in 2018 and 2019 are accessioned in the collections of the University of Alberta’s Laboratory for Vertebrate Paleontology (UALVP), Edmonton, Alberta, Canada.

Histology

For consistency, we followed the methods and definitions used in previous histological studies of hadrosaurids (Horner, Ricqles & Padian, 1999; Horner, De Ricqles & Padian, 2000; Vanderven, Burns & Currie, 2014; Woodward et al., 2015, Wosik et al., 2020). We sectioned the eight most complete humeri along the diaphysis (distal to the deltopectoral crest), as they offer the best sampled and most readily prepared bone from the bonebed (Fig. 4; Table 1). Humeri were chosen because they form the basis for MNI, guaranteeing that histological comparisons are performed on distinct individuals and approximating the relative age in the assemblage. Although humeri have been used in multiple hadrosaurid histological analyses (Horner, De Ricqles & Padian, 2000; Vanderven, Burns & Currie, 2014; Wosik et al., 2020), future studies could sample yet-to-be-prepared tibiae and femora, to better constrain the absolute ages of these hadrosaurids (Horner, Ricqles & Padian, 1999; Horner, De Ricqles & Padian, 2000; Vanderven, Burns & Currie, 2014; Woodward et al., 2015, Wosik et al., 2020).

Figure 4 Right (top) and left (bottom) humeri recovered from the Spring Creek Bonebed and denoting the minimum number of individuals (MNI = eight) and their consistent size.

Humeri show the typical lack of weathering and abrasion observed throughout the Spring Creek Bonebed. Additionally, humeri exhibit a variety of fracture styles and diagenetic distortion, causing the visible morphological variation. Note that letters correspond to Table 1. Humeri denoted with an asterisk were sectioned for histological analyses.

Table 1 List of humeri recovered from the Spring Creek Bonebed as seen in Fig. 4.

Figure identifier	Specimen number	Element	Length (mm)	Sectioning institution	
A	TMP 1991.137.0005	Right humerus (proximal)	282*		
B	UALVP 60537	Right humerus	261	UoA	
C	UALVP 60539	Right humerus	247	UoA	
D	UALVP 60534	Right humerus	246**	UoA	
E	UALVP 60536	Right humerus	262**	UoA	
F	UALVP 60532	Right humerus (distal)	268*	UoA	
G	UALVP 60535	Right humerus (proximal)	252*		
H	UALVP 60541	Right humerus (proximal)	270*		
I	TMP 1991.127.0001	Left humerus	253		
J	UALVP 60533	Left humerus (distal)	231*	UoA	
K	TMP 1991.137.0009	Left humerus (distal)	279*	UNE	
L	TMP 1988.94.0002	Left humerus (distal)	235*		
M	TMP 1988.94.0006	Left humerus (distal)	254*	UNE	
	Humerus length:	Mean = 257 mm	sd = 15.5 mm		
Notes:

* Estimated lengths.

** Underwent diagenetic modification.

Thin sections of TMP specimens were produced at the University of New England (UNE; Australia), and thin sections of UALVP specimens were produced at the University of Alberta (UoA; Canada). Humeri sectioned at the UNE were partially encased in epoxy resin to minimize damage during sectioning. The sections were cut using a diamond saw, before being mounted on slides and hand-polished with 600 grit silicon carbide. Slides were then placed into a Petrothin thin sectioning machine and ground down to 200 µm. The slides were then placed into a Logitech LP50 polisher to be ground down to 30 µm. Sections were analyzed under 10× magnification on a Leica DM500 compound microscope and were photographed under LED lighting with a Canon EOS 5DS.

Humeri prepared at the UoA were sectioned at mid-diaphysis using a table saw. Sections were then placed into plastic containers before being covered by EAGER Polymers’ EP4101UV Crystal Clear Polyester Resin (Castolite A.P. & Castolite A.C.) and EP4920 MEK-P Castolite Hardener (mixed in a 1 oz: 10 drops volume ratio). The cured resin blocks were cut in half using a table saw and mounted on plexiglass slides. Prior to mounting, both the plexiglass slides and resin blocks were faced using 1,000 grit silicon carbide grinding mixture. The sections were then ground down on a Hillquist saw using 600 and 1,000 grit grinding mixtures until suitable transparency, rather than any predefined thickness, was achieved. Images were captured under 4× magnification using a Nikon DS-FI3 camera, mounted on a Nikon Eclipse E600 POL microscope, and Nikon NIS Elements (v. 4.60) imaging software housed in the Caldwell Lab, UoA.

Taphonomy

All specimens were identified and inspected for taphonomic and preparation artefacts following laboratory preparation. Taphonomic analyses follow the procedures outlined by Behrensmeyer (1991) and built upon by Eberth, Rogers & Fiorillo (2007) and Blob & Badgley (2007). Taphonomic parameters were broadly categorized into either assemblage, quarry, or bone modification data, and analyzed under subcategories as outlined by Behrensmeyer (1991). In this study, a specimen is defined as a vertebrate hard part (e.g., bone, tooth, and scale) regardless of possible association with another bone (Blob & Badgley, 2007). Accordingly, multiple fused bones represent a single specimen, whereas unfused, but associated, bones (e.g., a string of vertebrae) are counted individually as distinct specimens. An element is defined as a vertebrate hard part in its entirety, such as a complete tibia as opposed to a distal piece of a tibia (Badgley, 1986; Blob & Badgley, 2007). A broken, but matchable element (e.g., a femur broken into four pieces, or distal and proximal ends of a femur) is regarded as a single specimen if the pieces can be reassembled. Analyses of taphonomic data utilized the total number of specimens (N), the number of identifiable specimens (NISP), and the number of prepared specimens (NPSP). The NISP is larger than the NPSP because few prepared specimens could not be identified. Except where noted, bone modification data are based on the NPSP, whereas assemblage and quarry data are based on the NISP. The minimum number of individuals (MNI) was determined by counting the most common unique skeletal elements (Blob & Badgley, 2007), which in the case of the SCBB were right humeri (Fig. 4; Table 1).

Voorhies (1969) groups are commonly used to assess skeletal representation and fluvial influence in bonebeds (Gangloff & Fiorillo, 2010; Bell & Campione, 2014; Evans et al., 2015). However, their application to large-bodied extinct taxa has been questioned (Eberth, Rogers & Fiorillo, 2007; Britt et al., 2009; Peterson, Joseph & Bigalke, 2013), as Voorhies groups do not account for bone completeness and disarticulation, particularly of the skull, prior to transportation. Additionally, Voorhies groups were originally used to examine the taphonomy of skeletally fused mammals rather than reptiles. Such factors can cause inaccurate element counts, leading to incorrect ratios between Voorhies groups and false inferences regarding fluvial influence, but elements can be counted more accurately by accounting for the lack of skeletal fusion in younger hadrosaurids (Horner & Currie, 1994). Moreover, the relative proportion of element representation is more informative than absolute counts (Gangloff & Fiorillo, 2010; Bell & Campione, 2014). For this study, Voorhies groups are based on inferred susceptibility to transport given a specimen’s size (as redefined by Scherzer & Varricchio, 2010; Varricchio, 1995), and the expected numbers of each element in a single hadrosaurid skeleton were derived from Horner, Weishampel & Forster (2004) and Bell & Campione (2014) (Table 2).

Table 2 Inventory and categorization of bones in a complete juvenile hadrosaurid skeleton and expected vs observed proportions of Voorhies groups from the Spring Creek Bonebed.

Voorhies group I	Voorhies group II	Voorhies group III	
Category	Element	Count	Category	Element	Count	Category	Element	Count	
Light cranial elements	Premaxillae	2	Pectoral elements	Sternal plates	2	Limb bones	Humeri	2	
	Nasals	2		Coracoids	2		Radii	2	
	Lacrimals	2		Scapulae	2		Ulnae	2	
	Jugals	2	Dense cranial elements	Maxillae	2		Femora	2	
	Quadratojugals	2		Dentaries	2		Tibiae	2	
	Postorbitals	2		Braincase	1		Fibulae	2	
	Surangulars	2	Tarsals and metapodials	Astragali	2	Pelvic elements	Ilia	2	
	Exoccipitals	2		Metatarsals	6				
	Hyoids	2		Metacarpals	6				
	Squamosals	2	Pelvic elements	Pubes	2				
	Quadrates	2							
	Frontals	2							
	Ectopterygoids	2							
Digital elements	Pedal phalanges	24							
	Manual phalanges	24							
Ribs	Dorsal ribs	36							
Vertebrae (including isolated centra)	Cervical	13							
	Dorsal	18							
	Caudal	50							
	Sacral	9							
Vertebral processes	Transverse processes	84							
	Neural spines	49							
	Haemal arches	35							
Pelvic elements	Ischia	2		
	Expected proportion	90%	Expected proportion	6.6%	Expected proportion	3.4%	
	Observed proportion	48.1%	Observed proportion	17.3%	Observed proportion	34.6%	
Chi-squared results: X-squared = 43.12, df = 2, p-value << 0.001	
Note:

Expected proportions were adapted from a variety of sources (Varricchio, 1995; Horner, Weishampel & Forster, 2004; Scherzer & Varricchio, 2010; Bell & Campione, 2014). Observed proportions were calculated from the number of identifiable specimens.

Age class designation follows two criteria. The first is that of Horner, De Ricqles & Padian (2000), who identified six classes based on histology: early and late nestling, early and late juvenile, sub-adult, and adult. When histological data are not available, we follow the size-based criterion used by Evans (2010), in which individuals beyond perinatal size and yet to attain 50% of adult size are defined as juvenile.

Results

Anatomical descriptions

The most diagnostic elements recovered from the SCBB include a premaxilla, maxilla, and postorbital, all of which show unambiguous lambeosaurine affinities. These elements are described in detail below.

Premaxilla—A partial left premaxilla (UALVP 60537) is preserved in two equal-length but non-contiguous pieces, separated by a gap of several millimetres (Fig. 5). Most of the anterior region of the premaxilla is intact, revealing the facial angle and the shape of the bill. However, the posterior contact with the nasal is absent, as is most of the premaxillary contribution to the cranial crest.

Figure 5 Left lambeosaurine premaxilla (UALVP 60537) from the Spring Creek Bonebed.

(A) Lateral view, including life reconstruction based on Hypacrosaurus stebingeri (Brink et al., 2011). Grey region indicates the portion of the premaxilla that was preserved. (B) Dorsal view with a dashed white line indicating the perimeter of the exposed bony naris. Abbreviations: bn, bony naris; cdp, caudodorsal process; nv, nasal vestibule; om, oral margin.

The oral margin of the premaxilla is rugose and was likely covered by a keratinous rhamphotheca in life (Morris, 1970; Horner, Weishampel & Forster, 2004; Farke et al., 2013). In dorsal view, the oral margin is transverse anteriorly and broadly arcuate more posteriorly with a smooth transition to the post-oral region of the premaxilla. As a result, it does not form a distinct, ventrolaterally directed tab-like process, as seen in other juvenile lambeosaurines (such as Hypacrosaurus, Parasaurolophus, and Velafrons coahuilensis; Gates et al., 2007; Evans, 2010; Brink et al., 2011), although the development of this process varies ontogenetically in lambeosaurines (Table 3; Evans, 2010). The anterior third of the preserved length of the premaxilla’s dorsal surface is concave mediolaterally, corresponding to the contour of the bony naris. The preserved bony naris has a length:width ratio of 3.5, exceeding the ratio observed in the southern Laramidian taxa Magnapaulia laticaudus and V. coahuilensis (1.85–2.85; Prieto-Márquez, Chiappe & Joshi, 2012), though this is likely due to ontogeny (Prieto-Márquez, Chiappe & Joshi, 2012). The bony naris attenuates anterior to the crest-snout angle, similar to juvenile Lambeosaurus lambei, V. coahuilensis and Hypacrosaurus altispinus (Lull & Wright, 1942; Ostrom, 1961; Evans, Forster & Reisz, 2005; Gates et al., 2007; Evans, 2010), but in contrast to the more posterior attenuated bony naris seen in juvenile Corythosaurus (Table 3; Evans, Forster & Reisz, 2005; Evans, 2010).

Table 3 Comparisons between lambeosaurine cranial bones founds at the Spring Creek Bonebed and other juvenile lambeosaurines.

Spring Creek Lambeosaurine
Morphologies	Corythosaurus casuarius	Hypacrosaurus altispinus	Hypacrosaurus stebingeri	Kazaklambia convincens	Lambeosaurus sp.	Parasaurolophus sp.	Velafrons coahuliensis	
Premaxilla								
No ventrolateral tab-like process	NA	No	No	NA	Yes	No	No	
Bony naris attenuates anterior to crest snout angle	No	Yes	No	NA	Yes	NA	Yes	
Crest-snout angle ~158°	~155°	~169°	~150°*	NA	~156°	~162°*	~157°	
Maxilla								
Dorsal opening foramen	NA	Yes	NA	NA	NA	NA	Yes	
Secondary dorsal foramen	Yes	NA	NA	NA	NA	NA	NA	
Anterodorsal angle ~151°	~149°	~147°	~143°*	NA	~154°	~165°*	~144°	
Postorbital								
No prefrontal contact doming	Yes	Yes	Yes	No	Yes	Yes	Yes	
Arcuate antorbital
fenestra margin	Yes	No	Yes	No	No	No	No	
Bifurcated squamosal process	Yes	No	Yes	Yes	Yes	Yes	Yes	
Notes:

* Measured from reconstructions.

‘NA’ means that the region in question is not preserved in the comparative specimen(s). Morphological comparisons are based on the following specimens: Corythosaurus casuarius: ROM 759 from (Evans, Forster & Reisz, 2005); Hypacrosaurus altispinus: CMN 2247 from (Brink et al., 2011); Hypacrosaurus stebingeri: TMP.1994.385.01, TCMI 2001.96.02, and NSM-PV 20377 from (Evans, 2010); Kazaklambia convincens: PIN2230/1 from Bell & Brink (2013); Lambeosaurus sp.: ROM 758 from (Brink et al., 2011); Parasaurolophus sp.: RAM 14000 from (Farke et al., 2013); Velafrons coahuliensis: CPC-59 from (Gates et al., 2007).

The posterolateral process is missing, exposing part of the narial vestibule in lateral view. In lateral aspect, the posterodorsal process becomes more dorsally inclined posteriorly, in the region representing the anterior part of the base of the crest. As preserved, the posterolateral process suggests a crest-snout angle of ~158° (Table 3), which is closest to the angles mesasured from juvenile Parasaurolophus sp. (162°; RAM 14000; Farke et al., 2013), V. coahuliensis (157°; CPC-59; Gates et al., 2007), and H. altispinus (163°; CMN 2247; Evans, 2010). Additionally the crest-snout angle of the Spring Creek maxilla falls out of the ranges measured from Corythosaurus casuarius: 116–155°, Hypacrosaurus stebingeri: 140°–150°, and Lambeosaurus sp. 62–156° (Evans, Forster & Reisz, 2005; Evans, 2010; Brink et al., 2011, 2014).

Maxilla—The right maxilla (UALVP 59881b) retains the typical triangular body seen in all hadrosaurids (Horner, Weishampel & Forster, 2004; Evans, 2010; Brink et al., 2011), despite lacking most of the dorsal process and roughly half of the maxillary body anterior to the dorsal process (Fig. 6A). The anterior fracture represents a dorsoventral shear revealing a cross-section of the most anteriorly preserved maxillary tooth family, showing at least three replacement teeth enclosed within the maxillary body (Fig. 6E). A shelf that would have supported the posterolateral process of the premaxilla extends medially from the maxillary body (Figs. 6C, 6D and 6E; Horner, Weishampel & Forster, 2004). Lateral to the medial shelf is a large dorsal foramen that opens along the anterodorsal margin of the dorsal process (Fig. 6C); the presence of a foramen at this location is characteristic of lambeosaurines (Horner, Weishampel & Forster, 2004). Lateral to the large dorsal foramen is a smaller foramen (Fig. 6C), as also seen in juvenile C. casuarius (ROM 759: Evans, Forster & Reisz, 2005).

Figure 6 Right lambeosaurine maxilla (UALVP 59881b) from the Spring Creek Bonebed.

(A) Lateral view, showing hypothetical reconstruction based on Hypacrosaurus sp. (MOR 553s). The dashed white line indicates the anterior margin of the sutural surface for the jugal. Black arrows indicate the location of lateral foramina. (B) Medial view. (C) Anterodorsal view. (D) Dorsal view. (E) Anterior cross section and schematic showing three internal teeth and one erupted tooth. (F) Ventral view. Abbreviations: af, alveolar foramina; df, dorsal foramen; dp, dorsal process; ec, ectopterygoid ridge; mt, maxillary teeth; ps, premaxillary shelf; sdf, secondary dorsal foramen; ssj, sutural surface for the jugal.

In the lateral aspect, the preserved portion of the dorsal process extends dorsally, forming an angle of ~151° with the anterodorsal edge of the maxillary body (Fig. 6A; Table 3). This angle is similar to that seen in juvenile C. casuarius (e.g., ROM 759), L. lambei (e.g., ROM 758), and V. coahuilensis (Gates et al., 2007), but differs from the more obtuse angles seen in subadult H. stebingeri (TMP 1994.385.0001: Brink et al., 2011), juvenile H. altispinus (CMN 2247: Evans, 2010), and juvenile Parasaurolophus (RAM 14000: Farke et al., 2013). The lateral aspect of the dorsal process is mostly occupied by the sutural surface for the jugal, which is anteriorly delimited by a distinct, roughly arcuate ridge. The shape of the ridge indicates that the anterior process of the jugal was broadly rounded, as in most lambeosaurines (Lull & Wright, 1942; Evans, Forster & Reisz, 2005; Evans, 2010; Brink et al., 2011), rather than distinctly pointed, as typically observed in hadrosaurines and Parasaurolophus (Horner, 1983, 1992; Prieto-Márquez & Norrell, 2010; Bell, 2011a; Prieto-Márquez, 2012; Xing, Mallon & Currie, 2017).

The ectopterygoid ridge projects from the maxilla laterally at a level ventral to the contact surface for the jugal and extends anteroposteriorly along the posterior two-thirds of the preserved maxillary body. In the lateral aspect, the ridge is mostly parallel to the tooth row but is deflected ventrally at the posterior end. In dorsal view and posterior to the dorsal process, the ectopterygoid ridge forms a mediolaterally broad shelf. Viewed posteriorly, the lateral margin of the shelf forms a lip curving ventrally, similar to Parasaurolophus sp. (RAM 14000: Farke et al., 2013), M. laticaudus (Prieto-Márquez, Chiappe & Joshi, 2012), and H. altispinus (CMN 8675: Evans, 2010). The ectopterygoid ridge partially covers the posteriormost foramen of a series of three foramina piercing the lateral surface of the maxillary body. Although these foramina consistently occur in the same general area in lambeosaurines, the specific number, shape, and position of these foramina are subject to individual and ontogenetic variation (Evans, 2010).

The nearly horizontal maxillary tooth row extends anteroposteriorly along the entire preserved length of the maxilla. The incomplete tooth row includes 23 identifiable tooth families, which alternate between one or two functional teeth on the occlusal surface (Fig. 6F). The number of functional teeth per tooth family ranges from one to three in hadrosaurids (Horner, Weishampel & Forster, 2004).

Postorbital—The nearly complete left postorbital (UALVP 59902) is triradiate in lateral view, typical for hadrosaurids (Lull & Wright, 1942). Three major processes are preserved: the anterior process, anteroventrally oriented jugal process, and posteriorly oriented squamosal process. A smaller medial process is also preserved (Fig. 7). The postorbital has undergone evident diagenetic distortion, the dorsal and lateral surfaces having been flattened into one plane. In connection with this, there is a large depression on the dorsal surface of the anterior process and a corresponding sinuous crack on the ventral surface, although such a crack may represent a groove for nerves and blood vessels.

Figure 7 Left lambeosaurine postorbital (UALVP 59902) from the Spring Creek Bonebed.

(A) Dorsolateral view. (B) Ventromedial view. White dashed line outlines the shape of the latersphenoid fossa. Note the longitudinal fracture on the medial surface in (B), which could also represent a neurovascular canal. Abbreviations: jp, jugal process; or, orbital rim; sp, squamosal process; ssf, sutural surface for frontal; ssp, sutural surface for parietal; sspf, sutural surface for prefrontal.

The anterior process is broad mediolaterally and triangular, with a deeply interdigitated sutural surface for the prefrontal along its anteromedial margin. There are no signs of doming at the prefrontal sutural surface (Table 3), unlike Kazaklambia convincens, where a prominent postorbital dome occurs on the dorsal surface of the bone (Bell & Brink, 2013). The prefrontal sutural surface terminates posteriorly at a small medial process, marking the separation between the prefrontal and the frontal sutural surfaces. Accordingly, the frontal was excluded from the orbital margin, as is typical for lambeosaurines (Horner, Weishampel & Forster, 2004). The medial process is dorsoventrally broad at its base and tapers medially, suggesting that it underlay the prefrontal and frontal contact and was thus not visible in dorsal view. Posterior to the medial process, the sutural surface for the frontal is less interdigitated than that for the prefrontal and bears a longitudinally oriented groove that opens dorsomedially. In dorsal view, the frontal sutural surface is distinctly concave, owing to the aforementioned medial process combined with a more posteriorly positioned one that would have extended medially to contact the parietal (Horner, 1992; Evans, Forster & Reisz, 2005; Evans, 2010). The ventral margin of the anterior process of the postorbital and the anterior margin of the jugal process form the posterodorsal rim of the orbit. The orbital rim is slightly rugose and is not pierced by a foramen, present in Amurosaurus riabinini, Prosaurolophus maximus, and Maiasaura peeblesorum (Horner, 1983, 1992; Godefroit, Bolotsky & Van Itterbeeck, 2004).

The jugal process is damaged at its midpoint, resulting in an unnatural anterior deflection of the ventral end. The lateral surface of the jugal process is concave, as seen in the juvenile C. casuarius (AMNH 5461), and is broader anteroposteriorly than that of Parasaurolophus, though not to the extent seen in Edmontosaurus (Parks, 1922; Wiman, 1931; Ostrom, 1961, 1963; Campione & Evans, 2011). The medial surface of the jugal process bears a prominent dorsoventral ridge that bifurcates dorsomedially to form a V-shaped fossa for the dorsolateral process of the laterosphenoid. This fossa is typically hemispherical/semicircular in hadrosaurids (e.g., Prosaurolophus maximus (MOR 447 6.24.6.2), E. regalis (ROM 53513 and 53514), Brachylophosaurus canadensis (MOR 1071 6.30.89.4), and L. magnicristatus (Evans & Reisz, 2007)).

The squamosal process is the longest of the postorbital processes. The ventral margin of the process forms the dorsal rim of the infratemporal fenestra, and is arcuate (Table 3), similar to that seen in juvenile H. stebingeri (TMP 1994.385.01; Brink et al., 2011) and C. casuarius (ROM 759; Evans, Forster & Reisz, 2005), but unlike the straight margins seen in juvenile Parasaurolophus sp. (RAM 14000; Farke et al., 2013), K. convincens (PIN 2230/1; Bell & Brink, 2013), and Lambeosaurus sp. (ROM 758; Brink et al., 2011). The squamosal process is also unlike the autapomorphic “dorsally positioned, high arching squamosal process” seen in V. coahuilensis (CPC-59; Gates et al., 2007). The posterior end of the process is lateromedially expanded, as in other lambeosaurines (Lull & Wright, 1942; Farke et al., 2013), and bifurcated, as in all lambeosaurines except H. altispinus (Evans, 2010). The ventromedial surface of the squamosal process bears an anteroposteriorly oriented groove, and the area lateral to the groove is broader and more prominent than that medial to the groove.

Histology

A general pattern of bone microstructure is present across the eight sampled humeri. The humeri comprise a thick layer of cortical bone externally, and a core of trabecular bone positioned at the centre of the diaphysis. No humeri exhibit a hollow medullary cavity. The trabeculae consist of parallel-fibred bone, although the deepest part of the core of trabecular bone is destroyed in most specimens due to diagenetic modification. External to the inner cancellous bone, most sections show regions of dense Haversian systems that have replaced the primary bone matrix (Fig. 8).

Figure 8 Thin sections of Spring Creek Bonebed humeri showing bone microstructure.

(A) Thin section of a humerus (UALVP 60539) showing the typical bone microstructure of humeri from the Spring Creek Bonebed, as described in the text. Scale bar = 1 mm. White arrows: (1) cancellous bone; (2) reticular bone; (3) plexiform bone; (4) laminar bone; (5) Haversian bone. (B) Laminar bone from UALVP 60533. Scale bar = 500 µm. (C) Reticular bone from UALVP 60535. Scale bar = 500 µm. (D) Plexiform bone from UALVP 60535. Scale bar = 500 µm. (E) Haversian reconstruction from UALVP 60539. Scale bar = 500 µm.

The outer half of the cortex comprises woven-fibred bone that ranges from plexiform to reticulate, transitioning to laminar bone towards the periosteal surface. Open osteonal canals are sporadically present on the periosteal surfaces of the humeri; we identified no external fundamental systems. Resorption fronts are present in all sections. Regions of Haversian reconstruction are restricted to the innermost layers and do not appear within the outer laminar layer. Neither annuli nor lines of arrested growth (LAGs) were observed in any of the sections. All aspects of bone microstructure indicate that skeletal growth was incomplete at the time of death.

Taphonomy

Assemblage data—A total of N = 351 vertebrate specimens were collected from the SCBB, including partial and complete teeth, ossified tendons, and bones. The NISP is 273, of which NPSP = 142. Almost all (99.7%) of the identifiable specimens are assigned to Hadrosauridae, with only a single tyrannosaurid tooth as direct evidence of a second taxon within the bonebed, although toothmarks suggest other taxa were present before deposition. The three cranial elements described above can be referred to Lambeosaurinae. Based on the number of right humeri collected (Fig. 4, Table 1), the current MNI of hadrosaurids is eight.

The maximum preserved lengths of individual specimens range from 10–640 mm (mean = 166 mm; median = 136 mm). Specimen lengths are positively skewed (skewness = 1.4918), the vast majority of elements being <400 mm in total length. Complete examples of each type of element tend to be uniform in size, indicating the occurrence of a single growth stage, which is supported by the histological results. Complete femora range from 558 mm to 640 mm, placing them in the late juvenile growth stage (sensu Horner, De Ricqles & Padian, 2000). Total lengths of postcranial elements, scaled against a complete Lambeosaurus sp. (AMNH 5340) skeleton (Farke et al., 2013), suggest a total body length estimate of ≤4.3 m (Table 4).

Table 4 Postcranial elements from the Spring Creek lambeosaurine scaled against elements from a complete Lambeosaurus sp. (AMNH 5340; from Farke et al., 2013) to estimate total body length.

Taxon	Lambeosaurus sp. (Farke et al., 2013)	Lambeosaurinae indet. (this study)	Ratio of Spring Creek Specimens to AMNH 5340	
Specimen	AMNH 5340	Longest Spring Creek specimens		
Humerus length (mm)	305	261 UALVP 60537	0.86	
Femur length (mm)	590	558 BADP 2019.0813.03**	0.94	
Tibia length (mm)	550	547 TMP.1995.024.0002	0.99	
Fibula length (mm)	530	455 TMP.1991.137.17	0.86	
Total Length (m)	4.31	≤4.27*		
Notes:

* Estimated body length.

** Based on field measurements of unprepared specimens.

Brackets indicate scale factor from AMNH 5340.

The vast majority of the bones were found disarticulated, with limited signs of association. The only possible exception pertains to a dentary, a predentary, and a mass of articulated teeth that were all found within an area <0.5 m2 (Fig. 2). The representation of Voorhies groups in the SCBB sample is more uniform than expected, given the juvenile hadrosaurid skeleton structure (Table 2; X2 = 43.12, p-value ≪ 0.001). In particular, there is a significant underrepresentation of Voorhies group I relative to groups II and III.

Quarry Data—The lateral extent of the 2018 and 2019 excavations was approximately 18 m, and the total excavation area was 35 m2 (Fig. 2). A femur recovered ~15 m upstream from the main excavation site, of similar size and preservation style to those recovered from the quarry, suggests a possible lateral extent of up to 33 m for the SCBB. The fossiliferous horizon is limited to the bottom 40 cm of a ~2 m thick mudstone layer with no distinct grading of bioclasts. Crevices (10–15 cm wide) found within the quarry walls suggest widespread slumping and the possible displacement of the entire bonebed from its original position. The density of bones within each grid square ranges from 1 to 30 bones/m2, with a mean of 7.5 bones/m2. Preferential alignment of long bones was difficult to determine from the SCBB as Rao’s spacing test suggests a significant departure from uniformity (test statistic = 183.6; critical value (at p = 0.05) = 143.8), whilst Kuiper’s test of uniformity suggests a uniform distribution (test statistic = 1.05; critical value (at p = 0.05) = 1.7). This inconsistency between tests may be related to the high circular variance (σ2 = 0.93) caused by high bone orientation variability (Fig. 2). Patchiness indices >1 were recorded from both 2018 and 2019 excavations (1.67 and 1.35, respectively), suggesting clumping of specimens rather than a random distribution.

Bone Modification—Of the prepared specimens from the SCBB, 44.2% are complete (Table 5), ranging in size from small cranial elements to relatively large hindlimb elements. A mixture of transverse post-burial fractures and perimortem spiral fractures represents the most common fracture style, observed on 38.9% of the NPSP. Signs of abrasion are rare within the SCBB, with only 13.9% of the NPSP showing low-level abrasion (stages 0 and 1) and <2% exhibiting more severe levels (stages 2 and 3). Similarly, 89.7% of NPSP show little to no weathering (weathering stages 0–1; Table 5, Fig. 4). The remaining 10.3% were observed to be at weathering stage 2.

Table 5 Taphonomic observations from the Spring Creek Bonebed, including the results of Chi-squared tests on the number of prepared specimens.

Weathering stage
(Behrensmeyer, 1978; Fiorillo, 1988)	Observed proportion	Abrasion stage
(Fiorillo, 1988)	Observed proportion	Fracture style	Observed proportion	
Stage 0: no signs of cracking or flaking on bone.
Possible years exposed after death: 0–1	72.9%	Stage 0: bone is unabraded, preserving all processes and edges.	84.3%	Complete: bone is preserved in its entirety.	44.2%	
Stage 1: bone is beginning to show signs of longitudinal cracking.
Possible years exposed after death: 0–3	16.8%	Stage 1: slight abrasion with some rounding of edges.	13.9%	Spiral: fractures with irregular fracture surfaces preserved from pre-burial.	7.6%	
Stage 2: thin layers of bone flaking, typically associated with longitudinal cracks.
Possible years exposed after death: 2–6	10.3%	Stage 2: moderate abrasion in which edges are well-rounded, and processes may or may not be identifiable.	0.9%	Transverse: straight, transverse fractures preserved from post burial.	9.3%	
Stage 3: patches of exposed fibrous texture where concentrically layered bone has been removed.
Possible years exposed after death: 4–15+	0%	Stage 3: high level of abrasion, edges extremely rounded, original bone shape is barely recognisable.	0.9%	Mixed: both transverse and spiral fractures preserved.	38.9%	
Chi-squared results: X2 = 71.08
p-value ≪ 0.001	Chi-squared results: X2 = 191.77
p-value ≪ 0.001	Chi-squared results: X2 = 44.538
p-value ≪ 0.001	
Proportion of specimens observed with toothmarks: 3.8%	Proportion of specimens observed with parallel striae: 28.2%	

Biogenic modification of some bones in the sample can be inferred based on the presence of parallel striae, which result from bone–substrate interactions and imply trampling (Behrensmeyer, Gordon & Yanagi, 1986). Approximately 33% of the prepared specimens exhibit such striae. The aforementioned perimortem spiral fractures are also consistent with trampling. Tooth marks are present on 3.8% of NPSP and are represented by pits and conspicuous parallel score marks; these marks primarily occur on limb bones. Given that the tooth marks are predominantly small, U-shaped furrows (Fig. 9), it is likely that they were produced by small scavengers, potentially including small theropods (Bell & Campione, 2014; Bell & Currie, 2016). Some scavenging by larger theropods may have occurred based on the presence of a single shed tyrannosaurid tooth and toothmarks potentially left by smaller tyrannosaurid individuals. Finally, only a single notable pathology is present on the supra-acetabular process of an incomplete ilium (UALVP 60540, Fig. 9). The pathology comprises a hemispherical erosion of the lateral surface of the process, with smooth margins but an irregular and rugose internal surface.

Figure 9 Examples of bone modification from the Spring Creek Bonebed.

(A) Unhealed parallel toothmarks (white arrows) on the lateral surface of the left dentary (UALVP 59907) interpreted as post mortem scavenging. (B) Pathology (margin indicated by white arrows) on the lateral surface of the supra-acetabular process (sa) from an incomplete ilium (UALVP 60540). (C) Example of paralell striae on the diaphysis of a fibula (UALVP 59982).

Discussion

Taxonomy of the Spring Creek hadrosaurids

The Spring Creek hadrosaurids were preliminarily assigned to the hadrosaurid clade Hadrosaurinae (or Saurolophinae, sensu Prieto-Márquez, 2010) based on the low deltopectoral crests observed on the humeri (Tanke, 2004). However, the prominence of the crest is known to vary ontogenetically, especially among lambeosaurines (Egi & Weishampel, 2002; Horner, Weishampel & Forster, 2004), rendering this assignment questionable. The skull elements described in this study represent the first diagnostic cranial material from the SCBB and unequivocally support a lambeosaurine designation based on the following synapomorphies: external naris fully enclosed by the premaxilla, large oblate foramen opening dorsally on the anterodorsal margin of the maxilla, and jugal sutural surface on the maxilla with a broadly rounded anterior margin (Evans, 2010; Prieto-Márquez, 2010). Furthermore, the postorbital would have contacted the prefrontal, excluding the frontal from the orbital margin. This condition is typical of lambeosaurines, despite being present in the hadrosaurines Prosaurolophus and Saurolophus (Horner, 1992; Horner, Weishampel & Forster, 2004; Bell, 2011a, 2011b; McGarrity, Campione & Evans, 2013). The other specimens so far recovered from the SCBB are not diagnostic below Hadrosauridae. However, given their consistent size and the absence of conspicuous variations that could indicate the presence of multiple taxa, it is likely that all hadrosaurid specimens from the SCBB pertain to the same species.

Unfortunately, the available sample of disarticulated juvenile elements provides limited diagnostic information, making any taxonomic designation below Lambeosaurinae ambiguous. The relatively acute angle between the body and dorsal process of the maxilla (Fig. 6) is more consistent with that seen in C. casuarius and Lambeosaurus (Lull & Wright, 1942; Evans, Forster & Reisz, 2005) than that seen in Hypacrosaurus and Parasaurolophus (Evans, 2010; Brink et al., 2011; Farke et al., 2013). Similarly, the arched profile of the postorbital squamosal process is akin to that in C. casuarius and L. magnicristatus (Evans & Reisz, 2007). By contrast, the more anteriorly attenuated bony naris of the premaxilla (Fig. 5) resembles that of H. altispinus and Lambeosaurus (Lull & Wright, 1942; Ostrom, 1961; Evans, Forster & Reisz, 2005; Gates et al., 2007; Evans, 2010; Brink et al., 2011), and the relatively obtuse snout–crest angle of the premaxilla is most consistent with H. altispinus (Evans, 2010; Brink et al., 2014); but note that the postorbital squamosal process (UALVP 59902; Fig. 7) differs from that of H. altispinus in being bifurcated (Evans, 2010).

The SCBB lies within the upper strata of Unit 3 of the Wapiti Formation, which is roughly contemporaneous with the Bearpaw Formation and the Drumheller and Strathmore members of the lower Horseshoe Canyon Formation (Fig. 10). The SCBB lambeosaurines are, therefore, younger than known species of Corythosaurus, Lambeosaurus, and Parasaurolophus from the Dinosaur Park Formation and intermediate in age between H. stebingeri and H. altispinus from the Two Medicine and Horseshoe Canyon formations, respectively (Horner & Currie, 1994; Brink et al., 2011; Mallon et al., 2012; Eberth & Kamo, 2020). As a result, the SCBB is apparently not contemporaneous with any other hadrosaurid species known from Canada or the U.S.A. (Fig. 10), although it may be contemporaneous with the Mexican lambeosaurines V. coahuilensis (Cerro del Pueblo Formation; Gates et al., 2007) and M. laticaudatus (El Gallo Formation; Prieto-Márquez, Chiappe & Joshi, 2012).

Figure 10 Biostratigraphy and palaeobiogeography of temporally and spatially proximate Lambeosaurinae from Alberta, Canada, Montana, USA and Mexico.

(A) Biostratigraphic distribution of Lambeosaurinae across strata from northwestern (Fanti & Catuneanu, 2009; this study) and southern Alberta, Canada (Mallon et al., 2012; Eberth et al., 2013; Eberth & Kamo, 2020), Montana, USA (Horner & Currie, 1994; Campbell, Ryan & Anderson, 2019), western and northeastern Mexico (Lucas & Sullivan, 2006; Gates et al., 2007; Prieto-Márquez, Chiappe & Joshi, 2012; Fowler, 2017). The Wapiti and Horseshoe Canyon formations are subdivided into units and members, respectively. Grey strata represent marine formations. The dashed error ranges for the Spring Creek lambeosaurines represents a temporal range within Unit 3, between the Pipestone Creek Bonebed (~73.5 Ma; Currie, Langston & Tanke, 2008a) and the basal-most Horseshoe Canyon Formation (~74.4 Ma; Eberth & Braman, 2012). Paleobiogeographical distribution of Lambeosaurinae across Mexico (B), and Montana, USA, and Alberta, Canada (C) (Lull & Wright, 1942; Horner & Currie, 1994; Evans & Reisz, 2007; Evans, Reisz & Dupuis, 2007; Gates et al., 2007; Evans et al., 2009; Evans, 2010; Prieto-Márquez, Chiappe & Joshi, 2012). The silhouette of the Spring Creek lambeosaurine and Magnapaulia laticaudus were created by Scott Hartman and Dmitry Bogdanov, respectively. Both were vectorized by T. Michael Keesey and used under the creative commons attribution 3.0 unported license (https://creativecommons.org/licenses/by/3.0/). The remaining silhouettes were used and modified under the public domain dedication 1.0 license. All silhouettes were sourced from www.phylopic.org.

The SCBB is geographically located between the northernmost lambeosaurine in Alaska (Takasaki et al., 2019) and those from southern Alberta (Fig. 10). Moreover, the SCBB is at a far higher paleolatitude than the roughly contemporaneous Mexican lambeosaurine localities (Fig. 10). Faunal endemism was suggested for at least Unit 3 of the Wapiti Formation based on the occurrence of P. lakustai and B. certekorum, both of which are uniquely known from the Pipestone Creek Bonebed (Currie, Langston & Tanke, 2008a; Bell & Currie, 2016), although such endemism may be stratigraphic rather than biogeographic (Fowler, 2017). Additionally, the occurrence of E. regalis in Unit 4 may reflect a shift from endemism to dinosaur cosmopolitanism across Alberta (Bell et al., 2014a).

The fact that the SCBB specimens are geographically and/or stratigraphically isolated from all other documented lambeosaurine occurrences, combined with the potential rapid evolutionary turnover of lambeosaurines, as evinced from the Dinosaur Park Formation (Mallon et al., 2012), and the conflicting morphological signals described above, suggests that the lambeosaurine material from the SCBB may well represent a new species unique to the Wapiti Formation. Unfortunately, such a conclusion cannot be considered secure in the absence of more complete, especially more mature, cranial material that reveals a unique suite of character states. Irrespective of its precise taxonomic identification, the SCBB sample represents the first lambeosaurine material reported from the Wapiti Formation. The presence of a lambeosaurine in Unit 3 supports a similarity in overall faunal composition between portions of the Wapiti Formation in northwestern Alberta to those from the southeast (Fanti & Catuneanu, 2009; Mallon et al., 2012; Eberth et al., 2013; Fanti, Bell & Sissons, 2013; Fowler, 2017; Eberth & Kamo, 2020). Furthermore, this discovery supports the hypothesis that Late Cretaceous lambeosaurine distributions extend into high-latitude regions, recently suggested based on an isolated supraoccipital from the Prince Creek Formation of Alaska (Takasaki et al., 2019).

Taphonomy of the Spring Creek Bonebed

The SCBB is essentially monospecific, containing the remains of at least eight juvenile lambeosaurines (thus far represented by 350 hadrosaurid bones) and one tyrannosaurid (represented by a single shed tooth), which are inferred to have been buried in an organic-rich, quiet-water setting based on the bonebed’s mud-hosted facies. The tyrannosaurid tooth likely entered the assemblage via scavenging rather than through the same event that caused the death of the lambeosaurines, as non-dental tyrannosaurid material is yet to be recovered from the bonebed. Furthermore, the light to minimal weathering (Table 5; Fig. 4) indicates that all the bones remained exposed for about the same length of time (<12 months; as identified by Behrensmeyer (1978) and Fiorillo (1988)). Together, these observations suggest that the juvenile lambeosaurines perished in a mass mortality event, rather than through gradual attrition (Bell & Campione, 2014; Chiba et al., 2015; Funston et al., 2016; Ullmann et al., 2017).

The killing mechanism for the SCBB lambeosaurines remains unknown. The pathological lesion observed on a partial ilium (UALVP 60540; Fig. 9) resembles features resulting from Langerhans Cell Histiocytosis inferred in other hadrosaurids, based on its smooth margin and “wrinkled” internal surface (Rothschild et al., 2020). However, it is intuitively implausible that an osteologically borne disease instigated the mass mortality event. Coastal-plain flooding has been interpreted as the typical source of macrofossil bonebeds throughout the Upper Cretaceous of Alberta (Eberth, 2015). Like those hosting the SCBB, floodplain deposits are common within Unit 3 of the Wapiti Formation, attesting to periodic inundation while the formation was being deposited (Fanti & Catuneanu, 2009). However, the absence of aquatic vertebrates and the lack of either advanced hydraulic reworking or channel sediments indicate that the SCBB lambeosaurines did not drown within a channel (Bell & Campione, 2014). Additionally, the laminar bone deposited near the periosteal surface of sectioned humeri may indicate slower bone growth, suggesting that the SCBB lambeosaurines died during a cold or dry season (Wosik et al., 2020).

Following the mass mortality event, the lambeosaurine cadavers were exposed long enough for scavenging, trampling, and disarticulation to occur but were buried before substantial weathering could take effect. The ubiquitous disarticulation in the SCBB is most likely a product of skeletal immaturity, which sees juveniles disarticulating more rapidly than adults (Hill & Behrensmeyer, 1984; Horner & Currie, 1994). Scavenging and trampling, inferred from the presence of tooth marks, parallel striae, and spiral fractures, may have also contributed to disarticulation. However, scavenging processes were likely minor given the low occurrence of bite-marks (3.8%; Table 5) compared to other sites, such as the Danek Bonebed (30%; Bell & Campione, 2014), Bleriot Ferry Bonebed (~10%; Evans et al., 2015), and Scabby Butte Bonebed: Site 2 (6.2%; Campbell, Ryan & Anderson, 2019).

A significantly higher incidence of bones within Voorhies groups II and III at the SCBB (χ2 = 43.12, p-value ≪ 0.001; Table 2) indicates the selective removal of some smaller, more transportable elements. Presumably, fluvial factors were the primary sorting mechanism (Voorhies, 1969), although some small elements, including haemal arches and metacarpals, were preserved. Tooth marks and parallel striae suggest that scavenging and trampling, respectively, occurred at the SCBB but, given their low incidence, likely represented minor sorting roles compared to fluvial influences. The preservation of hadrosaurid teeth articulated within a dentary (UALVP 59900) is significant because the fragile lingual sheet of bone that keeps the teeth within the dentary is highly susceptible to post-mortem damage, indicating that the SCBB lambeosaurines were buried before such early deterioration could occur (Bell & Campione, 2014). Moreover, the scarcity of teeth within hadrosaurid bonebeds has been used to support a ‘bloat-and-float’ scenario (Gangloff & Fiorillo, 2010), during which teeth are lost as a result of hydraulic transport, following the loss of the thin lingual sheet. The presence of articulated and isolated teeth in the SCBB is inconsistent with this scenario and suggests little to no transport from the site of death. Although Rao’s spacing test indicates a significant NE–SW preferred orientation, the substantial circular variance around this modal orientation (Fig. 2) suggests a generally low fluvial influence on long bone alignment. Additionally, high patchiness indices and some skeletal associations suggest little reworking/transport of elements. Overall lack of abrasion in the sample (Table 5) also suggests limited transport (Hunt, 1978; Fiorillo, 1988), although the relationship between abrasion and transport can be highly variable (Behrensmeyer, 1982; Argast et al., 1987; Eaton, Kirkland & Doi, 1989). Given the above taphonomic evidence, we cannot unambiguously reject that some transport of elements occurred, and thus the SCBB can be conservatively regarded as a parautochthonous mass mortality bonebed.

Growth dynamics of the SCBB lambeosaurines

Based on their observed bone microstructure, the SCBB lambeosaurines were undergoing sustained, but not rapid, growth at their time of death (Horner & Currie, 1994; Horner, De Ricqles & Padian, 2000; Hubner, 2012). The regions of reticular to plexiform bone preserved in the deeper parts of the outer cortex indicate recent periods of rapid growth, whereas the presence of secondary osteons coupled with the increased organization of the laminar bone towards the periosteal surface suggest that individuals were experiencing a slower growth rate (Horner, Ricqles & Padian, 1999; Horner, De Ricqles & Padian, 2000; Huttenlocker, Woodward & Hall, 2013). Scaling of limb bones from the SCBB to those of an articulated juvenile Lambeosaurus indicates that the individuals had attained a total body length of ≤4.2 m (Table 4; Lambeosaurus data from Farke et al., 2013), which is around half the 7–10 m total body length observed in most adult hadrosaurids or a third of the total ~12 m length reached by giant hadrosaurids (Prieto-Márquez, Chiappe & Joshi, 2012; Hone et al., 2014).

Attempts to infer hadrosaurid growth strategies from histological analyses are inescapably convoluted, to say the least. In Maiasaura peeblesorum, Horner, De Ricqles & Padian (2000) identified six distinct ontogenetic stages based on bone microstructure patterns and the total lengths of associated femora: early and late nestling, early and late juvenile, sub-adult, and adult. The SCBB lambeosaurines bear the greatest histological resemblance to the late juvenile stage, as sectioned humeri display: (1) laminar, plexiform, and reticular bone, (2) Haversian reconstruction, including secondary osteons, (3) spongiose, but not hollow, marrow cavities, and (4) no evidence of LAGs or an external fundamental system (Horner, De Ricqles & Padian, 2000). Late juveniles are hypothesized to exhibit moderate to high growth rates and, based on bone diametral increases, should have reached the late juvenile stage 1.1–2.4 years after hatching (Horner, De Ricqles & Padian, 2000; Wosik et al., 2020). However, the SCBB lambeosaurines possibly had a different ontogenetic trajectory to that described for Maiasaura.

The lack of LAGs among the sampled SCBB humeri is consistent with a late juvenile designation. In M. peeblesorum, 0–1 LAGs were indicative of a late juvenile growth stage (Horner, De Ricqles & Padian, 2000). Vanderven, Burns & Currie (2014) demonstrated that, in E. regalis, LAGs occur more frequently in humeri than in femora of E. regalis, a pattern thought to reflect slower humeral growth. A single LAG was observed in the smallest E. regalis humerus, although this specimen was ~140 mm longer than the humeri collected from the SCBB. LAGs were previously interpreted as representing annual interruptions in growth (Horner, Ricqles & Padian, 1999; Horner, De Ricqles & Padian, 2000; Chinsamy et al., 2012; Vanderven, Burns & Currie, 2014; Woodward et al., 2015), and it is, therefore, possible that the lack of LAGs in the SCBB lambeosaurines means that they were not yet a year old at the time of death. However, given their considerable size, this is unlikely.

Alternatively, the lack of LAGs among the sampled SCBB humeri may be environmental rather than ontogenetic (Chinsamy et al., 2012; Vanderven, Burns & Currie, 2014). The Wapiti Formation represents a geographic transition between Alaska’s polar faunas and the more temperate zones of southern Alberta and northern Montana (Bell et al., 2014b). As such, the development of bone textural switches in Edmontosaurus sp. from Alaska could be the result of polar overwintering, with harsher seasons leading to growth interruption (Chinsamy et al., 2012), although distinct LAGs have also been noted in some hadrosaurids from temperate latitudes (Horner, Ricqles & Padian, 1999). Nevertheless, the lack of LAGs at the SCBB suggests that Unit 3 of the Wapiti Formation was deposited under relatively equable climatic conditions (Fanti & Miyashita, 2009). In any case, the use of LAGs to determine absolute age is evidently ambiguous, especially for humeri (Horner, Ricqles & Padian, 1999; Horner, De Ricqles & Padian, 2000; Vanderven, Burns & Currie, 2014; Woodward et al., 2015). We, therefore, adopt the more conservative approach of assigning the SCBB lambeosaurines to the late juvenile stage based on their degree of histological similarity with late juvenile individuals of M. peeblesorum (Horner, De Ricqles & Padian, 2000), and their overall body size.

Age segregation in hadrosaurids

Taphonomic data and the lack of adult or perinatal material indicate that the SCBB lambeosaurine material is best interpreted as the remains of a group of late juvenile individuals that perished in a single mass mortality event. Accordingly, the composition of the SCBB may reflect a demographic phenomenon known as age segregation—the aggregation and segregation of individuals of the same species based on age, typically in response to resource or spatial limitations (Rogers & Kidwell, 2007; Pelletier et al., 2016). Among dinosaurs, age segregation has been proposed as an explanation for juvenile-dominated bonebed samples of sauropods (Myers & Fiorillo, 2009), theropods (Raath, 1990; Currie, 1998; Zanno & Erickson, 2006; Varricchio et al., 2008), ceratopsians (Gilmore, 1917; Lehman, 2006; Mathews, Henderson & Williams, 2007; Zhao et al., 2014), thyreophorans (Galton, 1982; Jerzykiewicz et al., 1993; McWhinney, Matthias & Carpenter, 2004), and ornithopods, including lambeosaurines (Dodson, 1971; Norman, 1987; Forster, 1990; Varricchio & Horner, 1993; Scherzer & Varricchio, 2010; Eberth, 2015; Vila, Sellés & Brusatte, 2016; Wosik et al., 2020). Food is a limited resource in most ecosystems, and sympatric species often employ interspecific niche partitioning strategies to minimize the adverse effects of competition (Farlow, 1976; Du Toit & Cumming, 1999; Lehman, 2001; Mallon & Anderson, 2014). However, ecomorphological data have only distinguished major dinosaurian clades (e.g., ceratopsians vs. hadrosaurids vs. ankylosaurs), and it remains unclear how, and indeed whether, closely related species may have mitigated the effects of mutual competition (Mallon & Anderson, 2014). It is, therefore, possible that in dinosaurs, such as hadrosaurids, such mitigation was achieved via intra- rather than interspecific dynamics, with juveniles and adults partitioning food based on either dietary requirements and/or physiological capabilities. For instance, the fitness costs of dietary synchronization in sauropods (such as those associated with movement to new foraging areas and the need for more resting time) as a result of size difference between juveniles and adults were possibly eased by age segregation and age-based niche partitioning, a scenario supported by the existence of ontogenetically variable dental microwear patterns (Fiorillo, 1998; Myers & Fiorillo, 2009; Zhao et al., 2014; Pelletier et al., 2016). To date, the possibility of similar ontogenetic variation in dental microwear has not been investigated in hadrosaurids. However, younger hadrosaurids were clearly unable to reach the same maximum feeding heights as adults, implying that juveniles must have had a more restricted feeding envelope unless they were actively fed by mature individuals. Accordingly, the SCBB and other age-segregated bonebeds (Varricchio et al., 2008; Myers & Fiorillo, 2009; Scherzer & Varricchio, 2010; Eberth & Braman, 2012) may be a product of population-level resource partitioning strategies that mitigated competition between diverse communities of megaherbivorous dinosaurs (Mallon et al., 2012; Mallon & Anderson, 2013; Mallon et al., 2013; Mallon & Anderson, 2014).

An alternate explanation for age segregation at the SCBB, though one not mutually exclusive with resource partitioning, is that hadrosaurid life history and breeding strategies led to seasonal variation in the age structure among a population (Varricchio, 2011; Zhao et al., 2014). Aggregations of hadrosaurid nesting sites indicate colonial nesting behaviours in both lowland and upland areas (Horner, 1982; Tanke & Brett-Surman, 2001; Fanti & Miyashita, 2009). During nesting times, non-breeding individuals may have segregated away from the breeding population, being relatively large (~50% of typical adult size; Table 4), but potentially still sexually immature (Varricchio et al., 2008; Varricchio, 2011, Wosik et al., 2020). However, such segregated groups are liable to contain a spectrum of ages (e.g., early–late juveniles; Varricchio et al., 2008; Zhao et al., 2014), which is not the case for the SCBB lambeosaurines. Finally, age segregation could be the result of annually cyclical parental caring behaviours, in which young were reared for an extended period within a yearly cycle, as observed in most modern crocodilian and avian species (Thorbjarnarson & Hernandez, 1993; Davies, 2002). Such parental caring behaviours have been inferred from multiple dinosaur bonebeds, including those of hadrosaurids (Horner & Makela, 1979), and supported by egg-adult associations (Varricchio, 2011).

Dinosaurs exhibited complex life histories and behavioural flexibility (e.g. Myers & Fiorillo, 2009; Varricchio, 2011), and there is still much about their palaeoecology that we do not understand (Mallon & Anderson, 2013; Mallon et al., 2013; Mallon & Anderson, 2014). Moreover, as we cannot readily distinguish between males and females in the dinosaur fossil record, we cannot reject the possibility that age-segregated dinosaur bonebeds were sexually segregated as well (Myers & Fiorillo, 2009; Pelletier et al., 2016), as was implied by Tanke (2004) in his original report, referring to the assemblage as a ‘bachelor herd’. Regardless of whether sex-segregation was typical of juvenile hadrosaurid bonebeds, such deposits offer a wealth of insights into growth and social behaviour in these ubiquitous herbivores and will undoubtedly reward further research.

Conclusion

This study marks the first formal description of the Spring Creek Bonebed and the first definitive documentation of lambeosaurines from the Wapiti Formation, here preserved within Unit 3. A total of 351 specimens were thus far collected from the bonebed, from which we identified a minimum of eight juvenile individuals based on non-overlapping humeri. Interestingly, unique spatiotemporal and conflicting morphological signatures hint at the presence of a new lambeosaurine species within the formation. However, given their ontogenetic state and the difficulties associated with identifying even complete juvenile specimens to a genus or species (Evans, Forster & Reisz, 2005; Brink et al., 2014), we feel that a conservative indeterminate Lambeosaurinae designation is the most appropriate at this time.

The seemingly exclusive preservation of a single age class adds to our understanding of dinosaurian life histories, further supporting that breeding, seasonality, and/or dietary partitioning may contribute to dinosaur demographics. Future research into macrofossil bonebeds, particularly from the Wapiti Formation, will undoubtedly provide a greater understanding of dinosaur diversity, distribution, and life history strategies during the final stages of the Mesozoic.

Supplemental Information

Supplemental Information 1 Additional images of humeri thin sections from the Spring Creek Bonebed.

(A) UALVP 60534. (B) UALVP 60537. (C) UALVP 60536. Scale bars represent 500 µm.

Click here for additional data file.

Supplemental Information 2 Additional images of humeri thin sections from the Spring Creek Bonebed.

(A) UALVP 60533. (B) UALVP 60532. (C) TMP 1988.94.0006. (D) TMP 1991.137.0009. White scale bars in (A) and (B) represent 500 µm. Images (C) and (D) were provided by Russell Bicknell.

Click here for additional data file.

Supplemental Information 3 Additional images of humeri thin sections from the Spring Creek Bonebed.

(A) UALVP 60533. (B) UALVP 60539. (C) UALVP 60532.

Click here for additional data file.

Supplemental Information 4 Metadata of all specimens recovered from the Spring Creek Bonebed.

Includes specimens accessioned from the Royal Tyrrell Museum of Palaeonotogy and the University of Alberta.

Click here for additional data file.

This study was completed as part of BH’s Master of Science degree at the University of New England, Australia, supervised by NEC and PRB. We are deeply indebted to Darren Tanke (Royal Tyrell Museum of Paleontology) for his role in prospecting and collecting in the Grande Prairie Region with Grande Prairie Regional College, and for originally documenting the SCBB. Thanks are given to John Scannella and Amy Atwater (Museum of the Rockies), Kevin Seymour (Royal Ontario Museum), Brandon Strilisky and Rebecca Sanchez (Royal Tyrell Museum of Paleontology), and Cam Reed and Calla Scott (Philip J. Currie Dinosaur Museum) for access to and help with their respective collections. Special thanks to Malcolm Lambert for preparing histological samples of TMP specimens and to Russell Bicknell for capturing images used in Figs. S2C and S2D. Thanks also to Kirstin Brink, Elizabeth Freedman Fowler, and Ryuji Takasaki for their expert reviews that greatly improved the standard of this work. We are grateful to Grande Prairie Regional College, who facilitated fieldwork in the Peace Region of northwestern Alberta, and to the many volunteers during the 2018 and 2019 BADP expeditions.

Additional Information and Declarations

Competing Interests

Author Contributions

Data Availability

The authors declare that they have no competing interests.

Brayden Holland conceived and designed the experiments, performed the experiments, analyzed the data, prepared figures and/or tables, authored or reviewed drafts of the paper, and approved the final draft.

Phil R. Bell conceived and designed the experiments, performed the experiments, analyzed the data, prepared figures and/or tables, authored or reviewed drafts of the paper, and approved the final draft.

Federico Fanti conceived and designed the experiments, performed the experiments, analyzed the data, authored or reviewed drafts of the paper, and approved the final draft.

Samantha Hamilton performed the experiments, analyzed the data, prepared figures and/or tables, and approved the final draft.

Derek W. Larson conceived and designed the experiments, performed the experiments, prepared figures and/or tables, rediscovery of the site, and approved the final draft.

Robin Sissons conceived and designed the experiments, performed the experiments, prepared figures and/or tables, preparation of fossil material, and approved the final draft.

Corwin Sullivan conceived and designed the experiments, performed the experiments, analyzed the data, authored or reviewed drafts of the paper, and approved the final draft.

Matthew J. Vavrek conceived and designed the experiments, performed the experiments, prepared figures and/or tables, rediscovery of the site, and approved the final draft.

Yanyin Wang performed the experiments, analyzed the data, prepared figures and/or tables, authored or reviewed drafts of the paper, and approved the final draft.

Nicolás E. Campione conceived and designed the experiments, performed the experiments, analyzed the data, prepared figures and/or tables, authored or reviewed drafts of the paper, and approved the final draft.

The following information was supplied regarding data availability:

The raw data is available in the Supplemental Files.

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
