# Peer review of "Taphonomy and taxonomy of a juvenile lambeosaurine (Ornithischia: Hadrosauridae) bonebed from the late Campanian Wapiti Formation of northwestern Alberta, Canada"

_PeerJ, doi:10.7717/peerj.11290_

## Round 0.1 · original submission · Minor Revisions

The three reviewers have identified a number of relatively minor comments to address during revision, outlined below.

·

Basic reporting

The manuscript describes a new lambeosaurine bonebed from the Wapiti Formation of northwestern Alberta, Canada. The new report is very important to add new views on the distribution and evolution of the Late Cretaceous hadrosaurids. Therefore, it should definitely be published in PeerJ, once a few issues written below are solved. Please note that I’m not a taphonomy expert that most of my comments are on anatomy sections of the manuscript. I expect another reviewer could cover the taphonomy part.

Given that I’m not a native English speaker, I cannot tell if the language is professional. However, I find repetitions and redundant sentences throughout the manuscript. I commented a few (please check below), but I suggest the authors carefully go over the manuscript once again.

There are several non-relevant literature citings. I found some cases citing papers to support something the original papers did not intend or never mentioned. I commented on the ones I noticed (please check below), but I assume there must be more. I suggest the authors go back to the original papers very carefully once again.

The article structure is fine except that I didn’t see the Conclusions section, which I thought is a standard of PeerJ. Subheadings should be in bold and followed by a period. Tables are not numbered in the order they appear in the main text.

Figures are relevant except for Fig. 8. Please specify in Fig. 8A where Fig. 8B, C, D, and E come from. I also suggest including a whole section, at least as a supplement. Raw data is supplied.

Experimental design

The research is original and primary. The question is well defined. Methods are relevant and described in detail.

Validity of the findings

The taxonomic identification is done without selective reporting in my view. The histology needs some revision (please check below).

All underlying data is provided except that only one histological section is provided. I suggest providing a few more as a supplement to support the microstructure consistency.

Additional comments

General comments for the author
It is great to see a paper reporting a new lambeosaurine bonebed from Alberta. Here are a few line-by-line comments. The reference list is in the end.

55 – Better to add either (or both) Horner et al. (1999) and Horner et al. (2000) if mentioning growth rate. I believe Horner and Currie (1994) didn’t discuss much on growth rate.
150 – Please specify which Figure. Fig. 1?
184 – Please mention how many humeri from which sides were sectioned here. If there are any reasons why those specimens were chosen, please write that here.
240 – The table number does not correspond to the order it appears in the main text. Please re-order.
256-259 – I believe none of the cited papers actually says the feature is ontogenetically variable. Farke et al. (2013) instead mention that this feature is present in various ontogenetic stages. Please add more detail if there is original data that show ontogenetic differences in the extent of the process.
268-274 – Ontogenetic change should be mentioned here. A small Lambeosaurus (e.g., ROM 758) has a similarly angled premaxilla. Additional comparison with Parasaurolophus would also be helpful.
279 –A picture from an anterior view in Fig. 6 would be helpful.
282 – I’m assuming “a large oblate foramen” is equivalent to the “dorsal foramen” of Fig. 6. Please be consistent in terminology.
302 – A picture from a posterior view would be helpful. Features described in the texts should be visible in the figures as much as possible.
305 – Are these the lateral foramina arrowed in Fig. 6? They seem not covered by the ectopterygoid ridge. If not, please specify in the figure, which foramina are mentioned in the text.
313 – Three processes -> Three major processes. Since the authors spend several sentences on the medial process, the medial process can be introduced here instead.
319 – Incomplete sentence.
326 – Doesn’t make sense to me. The medial process must have a sutural surface with the frontal on its dorsal surface to get such inference.
333 – What are the “other taxa” mentioned here? I don’t know much about foramen on the postorbital, but I thought the foramen was more common in hadrosaurines than lambeosaurines. Also, Horner (1992) noted the number of foramina is subject to individual variations but I believe nothing about presence/absence was mentioned.
341 – Please also compare with lambeosaurines, since this paper describes lambeosaurine materials.
343 – Add closing bracket, please.
345 – I’m not understanding “transversely convex dorsal margin”. If you mean that the anterior and the squamosal processes form a wide-angle (rather than straight) in lateral view, that is a typical lambeosaurine feature.
352 – A picture of the whole thin-section and pictures from other sectioned samples as supplementary files would be helpful to confirm the description.
358 – I’m not a histology expert, but I believe “Haversian bone” is not a common word. Secondary osteon or Haversian system may be a better word.
364 – It is self-explanatory. Regions of Haversian reconstruction is another word for rich in secondary osteons. Please reword the sentence.
360 – 366 Everything in this paragraph suggests the sectioned specimens are young. So I suggest rewording the whole paragraph and bring the “…skeletal growth was incomplete at the time of death” to the end of the paragraph.
374 – In the description section, it was not clear enough which characters were used to refer the cranial materials to Lambeosaurinae. This may be due to the frequent use of “typical” or “most”, which sounds like there are exceptions. Adding a table that shows which characters were used to refer which materials may help.
381 – Large juvenile? Also, I prefer the “growth stage” rather than “age class”.
384 – Table 2, not Table 4. Also, the table does not show comparisons with Parasaurolophus.
384 – Which Spring Creek specimens were measured for Table 2? Are these the largest among the same elements? If so, the estimated total length will be 4.52 m or smaller and no minimum can be estimated from the data provided here.
384-387 – “Rapid bone growth” is mentioned here for the first time. The histology section only stated that skeletal growth was incomplete, without any citation. This last sentence seems redundant to me.
449 “mass mortality event” is mentioned here for the first time. I suggest deleting it or moving the whole section later to avoid confusion.
452 –Taxonomy could be narrowed down a little more. The premaxilla is certainly not of Parasaurolophini. The character distributions presented here makes me feel the SCBB lambeosaurine belong to Lambeosaurini.
452 – A table that summarized the sentences starting here would be helpful.
467 – As far as I know, Horner (1992) presents postorbital of Prosaurolophus but no other hadrosaurids, nor provide a comparative description. Also, I don’t know of any paper saying the laterosphenoid fossa shape may vary ontogenetically.
476 – Change hadrosaurid to lambeosaurine, please. Or, please provide comparisons with hadrosaurines, too.
492 – I agree that the SCBB lambeosaurine may represent a new species. Some comparisons with Asian lambeosaurines may support the claim. Recent phylogenetic hypotheses suggest close affinities of Asian taxa to Lambeosaurus (Kobayashi et al., 2019) or Hypacrosaurus (Prieto-Márquez et al., 2019).
515 – I’m not sure if these citations are relevant. This paragraph says the lambeosaurines died by a mass mortality event because the bonebed is monodominant and the bones experience minimal weathering. However, Scherzer & Varricchio (2010) described a bonebed with a larger degree of weathering and Campbell et al. (2019) described a multitaxic bonebed. I didn’t check all of the papers cited, and I’m not a taphonomy expert, but I feel there must be better papers that support the claim.
529 - Henderson (2014) never stated that hadrosaurs could swim. The paper said hadrosaurs could survive for hours to days in a flood. I also don’t think the paper mentioned anything about ontogeny.
568 – “Haversian reconstruction” is, as far as I know, not a common word. “Presence of secondary osteons” is enough.
571-576 – This part is repeating the result section. I’m also not sure why comparing body size with adult lambeosaurines of different taxa is important to discuss in this paper.
580 – Horner et al. (2000) recognized “small” and “large”, not “early” and “late”.
597 – I’m getting confused. The former paragraph stated the SBCC specimens are 1.1 ~ 2.4 years old and now saying they are less than a year old.
602 – According to Chinsamy et al. (2012), the polar Edmontosaurus does not show any LAGs, only vascular patterns.
603 – Please note that the Alaskan Edmontosaurus sp. is larger than the SBCC specimens (~65% adult size), although I agree that it has a large number of growth marks for its size.
608 – My understandings of Woodward et al. (2015) is that the vascular cycling pattern, not LAGs, is a response to seasonal (probably annual) hormonal shifts. Also, I personally think comparing humerus to tibia is not appropriate, since Fig. 8 shows a large medullary region compared to hadrosaurid tibia (e.g., Freedman Fowler & Horner, 2015; Woodward et al., 2015) that may obscure earlier growth marks.
616 – I agree that the use of LAGs for age estimation is not completely reliable, but I know that at least some of the papers cited here are not relevant to support the claim.
618 – This conclusion is a repeat of the second paragraph of this section. I suggest re-writing this whole section (“Growth dynamics of the SCBB lambeosaurines”). There are some redundancies, repetitions, and wrong use of citations. The subheading does not match the contents.
624 – Lack of perinatal material may be explainable by fluvial factors (as discussed in lines 540~).
651 – Yes, feeding height differences can result in a sympatric niche partitioning called browsing stratification (Arsenault & Owen-Smith, 2008; du Toit & Olff, 2014; Kleynhans et al., 2011). So I wonder if hadrosaur age segregation is for resource partitioning, since different body size seems to force different resource use.

References
Arsenault R, and Owen-Smith N. 2008. Resource partitioning by grass height among grazing ungulates does not follow body size relation. Oikos 117:1711-1717.
Campbell JA, Ryan MPJ, and Anderson J. 2019. A taphonomic analysis of a multi-taxic bonebed from the St. Mary River Formation (uppermost Campanian to lowermost Maastrichtian) of Alberta, dominated by cf. Edmontosaurus regalis (Ornithischia: Hadrosauridae), with significant remains of Pachyrhinosaurus canadensis (Ornithischia: Ceratopsidae). Canadian Journal of Earth Sciences.
Chinsamy A, Thomas DB, Tumarkin-Deratzian AR, and Fiorillo AR. 2012. Hadrosaurs were perennial polar residents. The Anatomical Record 295:610-614.
du Toit JT, and Olff H. 2014. Generalities in grazing and browsing ecology: using across-guild comparisons to control contingencies. Oecologia 174:1075-1083.
Farke AA, Chok DJ, Herrero A, Scolieri B, and Werning S. 2013. Ontogeny in the tube-crested dinosaur Parasaurolophus (Hadrosauridae) and heterochrony in hadrosaurids. PeerJ 1:e182.
Freedman Fowler EA, and Horner JR. 2015. A new brachylophosaurin hadrosaur (Dinosauria: Ornithischia) with an intermediate nasal crest from the Campanian Judith River Formation of northcentral Montana. PLoS ONE 10:e0141304.
Henderson DM. 2014. Duck Soup The Floating Fates of Hadrosaurs and Ceratopsians at Dinosaur Provincial Park. Hadrosaurs: Indiana University Press, 459-466.
Horner JR. 1992. Cranial morphology of Prosaurolophus (Ornithischia: Hadrosauridae) with descriptions of two new hadrosaurid species and an evaluation of hadrosaurid phylogenetic relationships. Museum of the Rockies Occasional Paper 2:1-119.
Horner JR, de Ricqlès A, and Padian K. 1999. Variation in dinosaur skeletochronology indicators: implications for age assessment and physiology. Paleobiology 25:295-304.
Horner JR, De Ricqlès A, and Padian K. 2000. Long bone histology of the hadrosaurid dinosaur Maiasaura peeblesorum: growth dynamics and physiology based on an ontogenetic series of skeletal elements. Journal of Vertebrate Paleontology 20:115-129.
Kleynhans EJ, Jolles AE, Bos MRE, and Olff H. 2011. Resource partitioning along multiple niche dimensions in differently sized African savanna grazers. Oikos 120:591-600.
Kobayashi Y, Nishimura T, Takasaki R, Chiba K, Fiorillo AR, Tanaka K, Chinzorig T, Sato T, and Sakurai K. 2019. A new hadrosaurine (Dinosauria: Hadrosauridae) from the marine deposits of the Late Cretaceous Hakobuchi Formation, Yezo Group, Japan. Sci Rep 9:12389.
Prieto-Márquez A, Wagner JR, and Lehman T. 2019. An unusual ‘shovel-billed’ dinosaur with trophic specializations from the early Campanian of Trans-Pecos Texas, and the ancestral hadrosaurian crest. Journal of Systematic Palaeontology:1-38.
Scherzer BA, and Varricchio DJ. 2010. Taphonomy of a Juvenile Lambeosaurine Bonebed from the Two Medicine Formation (Campanian) of Montana, United States. Palaios 25:780-795.
Woodward HN, Freedman Fowler EA, Farlow JO, and Horner JR. 2015. Maiasaura, a model organism for extinct vertebrate population biology: a large sample statistical assessment of growth dynamics and survivorship. Paleobiology 41:503-527.

·

Basic reporting

The paper is clear and well-written. The tables and supplementary information provide all necessary data. There are a few references that should be discussed/added to the manuscript, see comments below. There are also a few images that could be added to the figures, see comments below.

Experimental design

Methods are well described and executed. The bonebed and morphologically significant fossils are described with a good amount of detail.

Validity of the findings

Conclusions are well-supported by the data.

Additional comments

This is a well-written, detailed paper describing a lambeosaurine bonebed from the Wapiti Fm of Alberta. The figures are nice and the description of significant, identifiable lambeosaurine material is well done. I only have a few comments/suggestions:

-Figure 4 caption, state that letters correspond to letters in Table 1.

-Lines 278-280: Can you add in a photo of the maxilla in anterior view showing the tooth families? There is space for this in Figure 6.

-Line 319: sentence incomplete.

-Lines 313-350, post-orbital description. Can you comment on the curvature of the squamosal process that forms the dorsal margin of the lateral temporal fenestra? This curvature is more arcuate in some juvenile lambeosaurines (e.g. Kazaklambia, Bell and Brink 2013), and I’m wondering if this shape changes through ontogeny (which might be hard to determine at this point without new measurements). Also is there any evidence of bossing on the p-o, as in Kazaklambia, or a mound where the p-o meets the prefrontral (Brink et al 2011)? These characters could be potentially significant, since the p-o bossing has been described as an autapomorphy for Kazaklambia.

-Line 419: a large percentage of bones have striae (33%), can you add an example of these striae to Figure 10?

-Line 546, the preservation of teeth in jaws. Take a look at the Bramble et al 2017 (page 8) paper regarding the anatomy of the teeth in the jaws. Teeth fall out likely because of the soft tissue that holds the teeth together, not necessarily the thin wall of lingual bone.

-Line 660, the seasonal variation in cohort age structure. It would be worth discussing the results of the Wosik et al. 2020 paper here (and in other parts of the paper), which has an in-depth discussion about juvenile cohorts and size distributions in the Dinosaur Park Formation, as well as aging juvenile specimens based on the Horner et al (2000) Maiasaura scheme.

References

Bell PR, Brink KS. 2013. Kazaklambia convincens comb. nov., a primitive juvenile lambeosaurine from the Santonian of Kazakhstan. Cretaceous Research. 45:265-274.

Bramble K, LeBlanc ARH, Lamoureux DO, Wosik M, Currie PJ. 2017. Histological evidence for a dynamic dental battery in hadrosaurid dinosaurs. Scientific Reports. 7(1):15787.

Brink KS, Zelenitsky DK, Evans DC, Therrien F, Horner JR. 2011. A sub-adult skull of hypacrosaurus stebingeri (ornithischia: Lambeosaurinae): Anatomy and comparison. Historical Biology. 23(1):63-72.

Horner JR, Ricqlés AD, Padian K. 2000. Long bone histology of the hadrosaurid dinosaur Maiasaura peeblesorum: Growth dynamics and physiology based on an ontogenetic series of skeletal elements. Journal of Vertebrate Paleonotology. 20(1):115-129.

Wosik M, Chiba K, Therrien F, Evans DC. 2020. Testing size–frequency distributions as a method of ontogenetic aging: A life-history assessment of hadrosaurid dinosaurs from the dinosaur park formation of Alberta, Canada, with implications for hadrosaurid paleoecology. Paleobiology. 46(3):379-404.

·

Basic reporting

Well written & presented. See additional comments below.

Experimental design

Good, although see comment on humerus histology below.

Validity of the findings

Good.

Additional comments

Geological setting, line 158: “(Fig.)” is missing the number.

Methods- Histology:
I disagree with the choice of sectioning humeri rather than tibiae or femora (although I realize it’s obviously too late to change that). If this had been a bonebed of multiple sizes of individuals, and there were size classes represented by humeri but not tibiae or femora, then it might make sense. However, this is a bonebed remarkable for the consistent size of all individuals, so choosing the best bone for accurate histology results would have been preferable over sample size. In hadrosaurs, the tibia is the best to sample because it is a large weight-bearing bone that does not have large processes near the midshaft. In hadrosaurs, the femur has the fourth trochanter near the midshaft, which causes extra remodeling in that area. In the humerus, the large deltopectoral crest causes even more disruption. Sampling the shaft of the humerus requires deviating far from the mid-length of the bone, which means the early growth record will not be present (when the bone was shorter than the location of the slice).

Scanning the excel file of the specimen list, there are 10 femora plus 4 distal femora, and 12 tibiae plus 3 partial tibiae. So, in the manuscript, please explain in more detail why you chose to use the humeri instead of the tibiae and femora, which are the elements typically used in hadrosaur histological analyses.

How many of the 13 humeri were actually sectioned? Specimen H in figure 4 is far too fragmentary for useful histology.

How many histology specimens were processed at UNE vs UA?

Why were no images at all taken from the UNE sections? Even without a top-of-the-line automatic stage system, most universities have at least some sort of microscope camera-to-computer hookup. I’m not asking for the best photos, just some sort of data record.


Methods-Taphonomy-Voorhies groups: Why was “braincase” chosen to list as a single element? This depends on the degree of fusion/maturity of the specimens. Were any associated braincase elements found?

Description-Premaxilla-2nd paragraph: many comparisons are made with other “juveniles”, and ontogenetic changes are mentioned, but you don’t actually say how the bonebed material compares ontogenetically with these other specimens. Are they all the same size/stage? Did you already define how you are using the term “juvenile” here? I suggest the percent-of-adult-size definition of Evans (2010 ZJLS).

Description-Premaxilla-3rd paragraph: Are those crest-snout angles for adult specimens? Does this change ontogenetically? What are the angles in similarly sized juveniles of known species? (In the maxilla section that follows, examples neatly include the species, specimen number, and ontogenetic stage, which is very useful.)

Description-Postorbital-1st paragraph: Last sentence of paragraph isn’t finished.

Taphonomy-1st paragraph: Line 388 says “NSPS” instead of NPSP.

Discussion line 498: change “likely contemporaneous” to “may be contemporaneous”. Also, the correlations in the stratigraphic chart of Fowler 2017 suggest that Unit 3 may be distinctly below the positions of Velafrons and Magnapaulia, leaving the SCBB in a very unique temporal position.

Discussion-Faunal endemism-lines 504-506: Fowler 2017 suggests that P. lakustai’s uniqueness is stratigraphic, not biogeographic.

Needs separately labeled Conclusion paragraph.


Table 2: Which Spring Creek specimens were used? Are these numbers averages of available specimens? How many of each element type (n=)? Also, it is not easily clear that the numbers in parentheses represent the ratio compared to the AMNH specimen.

Figure 9: Part A is only useful if compared against the size distribution within a complete skeleton. Of course there are more small elements (vertebrae, phalanges) than there are large elements (femora, tibiae). Table 3 covers this already; 9A is not needed.

Figure 11A: In caption, first sentence, list regions in same order as they are in figure.

Figure 11B/C: Why do Velafrons and Hypacrosaurus have small letters behind their silhouettes? I don’t see those codes used anywhere.

Figure 11 figure credits: What was the source of the remaining public domain silhouettes? I assumed they would have been from T. Michael Keesey’s website.

Is there are Table Caption for the supp.info. metadata excel file? What do the dorsal/ventral and anterior/posterior columns mean?

---

## Round 0.2 · accepted · Accept

Thank you for your close attention to the comments from the reviewers.